# Effect of trade on global aquatic food consumption patterns

Kangshun Zhao [1,2], Steven D. Gaines [2], Jorge García Molinos [3], Min Zhang [4] ✉ & Jun Xu [1,5,6] ✉

Globalization of fishery products is playing a significant role in shaping the harvesting and use of aquatic foods, but a vigorous debate has focused on whether the trade is a driver of the inequitable distribution of aquatic foods. Here, we develop species-level mass balance and trophic level identification datasets for 174 countries and territories to analyze global aquatic food consumption patterns, trade characteristics, and impacts from 1976 to 2019. We find that per capita consumption of aquatic foods has increased significantly at the global scale, but the human aquatic food trophic level (HATL), i.e., the average trophic level of aquatic food items in the human diet, is declining (from 3.42 to 3.18) because of the considerable increase in low-trophic level aquaculture species output relative to that of capture fisheries since 1976. Moreover, our study finds that trade has contributed to increasing the availability and trophic level of aquatic foods in >60% of the world's countries. Trade has also reduced geographic differences in the HATL among countries over recent decades. We suggest that there are important opportunities to widen the current focus on productivity gains and economic outputs to a more equitable global distribution of aquatic foods.

The Sustainable Development Goals (SDGs) agenda puts food security and ending malnutrition as a global priority[1]. Aquatic systems have a significant role to play in meeting these objectives. Fisheries, aquaculture, and their trade are critical to the achievement of food security and sustainable economic, social, and environmental development goals[2,3]. In recent decades, global fisheries and aquaculture production have grown substantially. Aquatic foods are among the most highly traded commodities in the global food system and are becoming increasingly globalized[4,5].

As a highly diverse food group, aquatic foods are now widely recognized in global food systems and can supply critical nutrients and improve overall human health[6–9]. However, accelerating climate change, overfishing, industrial pollution, and coastal urbanization challenge the ocean's ability to meet growing aquatic food demands[10–13]. The percentage of fishery stocks at biologically unsustainable levels has increased from 10% in 1974 to 35.4% in 2019[14]. Promisingly, global aquaculture has rapidly developed over the past few decades and is thought to be the only reliable way to meet the growing future demand for aquatic foods[15,16]. Meanwhile, given the geographic patchiness of wild fish and aquaculture production, trade will be increasingly essential for the redistribution of global aquatic products and food security.

Our understanding of the wide diversity of aquatic species produced and traded worldwide, and the impacts of aquatic food

[1]Key Laboratory of Breeding Biotechnology and Sustainable Aquaculture, Key Laboratory of Lake and Watershed Science for Water Security, Institute of Hydrobiology, Chinese Academy of Sciences, Wuhan, China. [2]Bren School of Environmental Science & Management, University of California, Santa Barbara, CA, USA. [3]Arctic Research Center, Hokkaido University, Sapporo, Japan. [4]Hubei Provincial Engineering Laboratory for Pond Aquaculture, Engineering Research Center of Green Development for Conventional Aquatic Biological Industry in the Yangtze River Economic Belt, College of Fisheries, Huazhong Agricultural University, Wuhan, China. [5]State Key Laboratory of Marine Resource Utilization in South China Sea, Hainan University, Haikou, China. [6]Laboratory for Marine Fisheries Science and Food Production Processes, Qingdao National Laboratory for Marine Science and Technology, Qingdao, China. ✉e-mail: zhm7875@mail.hzau.edu.cn; xujun@ihb.ac.cn

products trade across geographies on food security goals remain ambiguous and the evidence is mixed[17-20]. While previous studies have provided essential insights into general aquatic food trade and consumption characteristics and trade impacts in several countries and regions[5,17,19-21], our collective understanding of the outcomes of aquatic food globalization is still limited by a fundamental gap between detailed production and trade data. Reconciling aquatic food production and trade data has remained a challenge due to mismatches in species- versus product-level reporting and weight losses during processing[22,23]. Compared with a deeper understanding of the role of trade in land-based food systems (e.g., agricultural and livestock products)[5,24,25], insights into global aquatic food consumption patterns and the impact of trade continue to lag far behind[26].

The trophic levels of animal or plant species, representing their relative positions in the aquatic food chains, are a primary metric used in ecological studies for a wide range of applications[27]. They not only represent a synthetic metric of species' diets, which is an important indicator of different aspects of the environmental footprint of food production for aquaculture and wild caught aquatic foods[28,29], but they are also widely recognized as an appropriate indicator of aquatic food value (i.e., higher trophic level generally corresponding to higher price)[30-32]. Although the trophic level of food items in the human diet (human trophic level, HTL) has been considered a simple composite metric that synthetically reflects global patterns of human diet[33], there is currently no quantitative assessment of the human aquatic food trophic level (HATL) and the impacts of trade on it. Nonetheless, we note that some small low-trophic level pelagic and inland fish are also nutrient-rich (e.g., calcium, iron, zinc, long-chain omega-3 polyunsaturated fatty acids)[6,8,34], and that wild-capture high-trophic level species are more likely to be contaminated with biomagnifying substances such as persistent organic pollutants (POPs), heavy metals, and microplastics[35-37]. Therefore, the trophic level of aquatic foods can

indicate the value of aquatic foods based on price, but it does not predictably reflect the concentration of any nutrients or contaminants status. Here, we first use the FAO national fisheries and aquaculture production and trade data (1979-2019) to develop a species-level mass balance dataset and a trophic level identification dataset for 174 countries and territories (hereafter called countries). We then calculate the HATL and per capita consumption across different countries and regions to analyze global aquatic food (i.e., fish, cephalopods, and crustaceans) consumption patterns, trade characteristics, and impacts.

## Results

### Aquatic food consumption patterns

Population, income growth, and associated changes in dietary habits are the main influential drivers of the increase in global fish demand in recent decades[38,39]. From 1976 to 2019, global per capita consumption of aquatic foods has increased significantly, but the HATL has declined from 3.42 to 3.18 (Fig. 1a), contrary to the global trend of HTL that also includes land based foods[33]. This declining trend for aquatic foods can be primarily explained through a combination of two factors. First, the global aquaculture trophic level is significantly lower than that of capture fisheries (nearly 0.8 lower than capture fisheries on average) (Fig. 1b). Second, while global capture fisheries production has experienced only a marginal increase in recent decades, aquaculture output has experienced a sustained and very rapid increase over the entire study period, especially in Asia (Fig. 1b, Supplementary Figs. 1 and 2). The proportion of aquatic foods originating from aquaculture production rose from 6% in the 1960s to 56% in 2020[14]. This finding indicates that aquaculture is driving the decreasing trend in global HATL, despite increasing consumption of aquatic food (Fig. 1a, b) and the rapid growth in the production of high-trophic level species driven by globalized trade and favorable economic conditions for large-scale

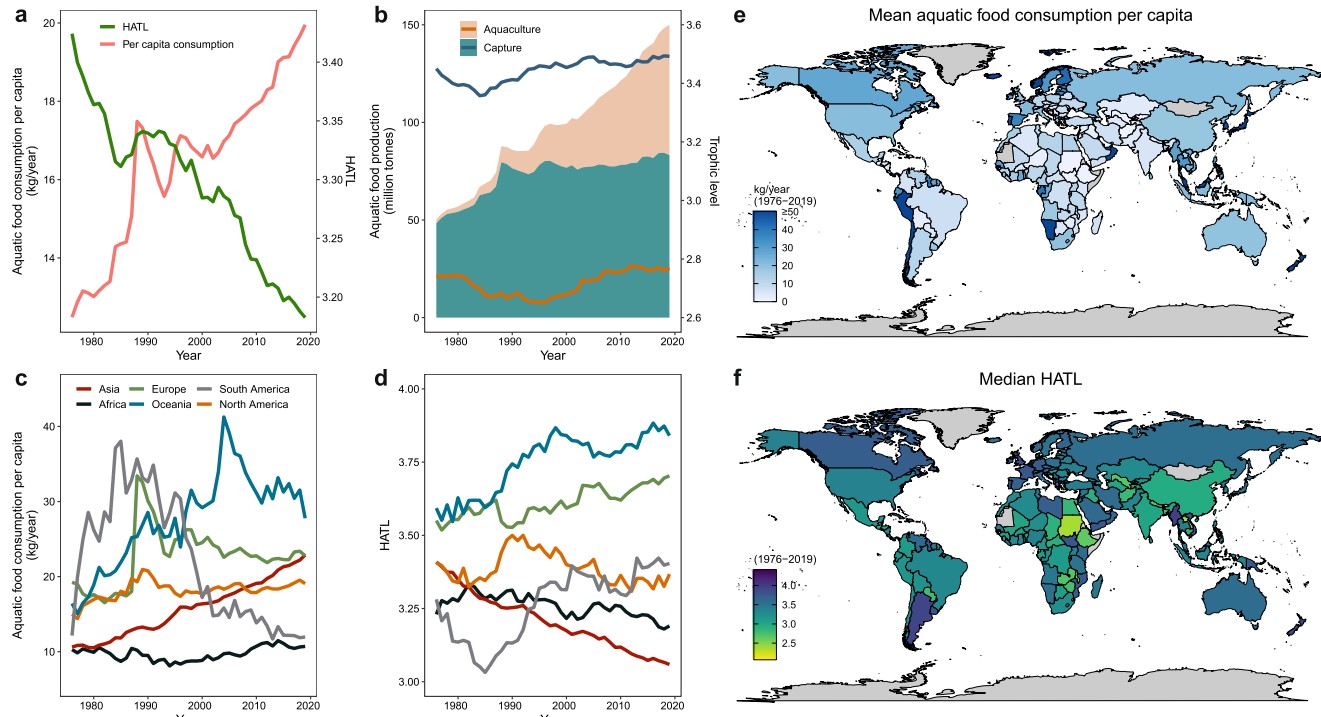

**Fig. 1 | Global trends of aquatic food production and apparent consumption from 1976 to 2019. a** The global change in per capita consumption of aquatic foods (inc. fish, cephalopods, and crustaceans) and HATL. **b** The global change in production and trophic level of aquaculture and capture fisheries. The line refers to trophic level, and the envelope refers to production. **c** Trends of per capita

consumption of aquatic foods in different continents. **d** Trends of HATL in different continents. **e** The mean country-level per capita consumption of aquatic foods. **f** The median country-level HATL. HATL human aquatic food trophic level. Countries in gray: No data available.

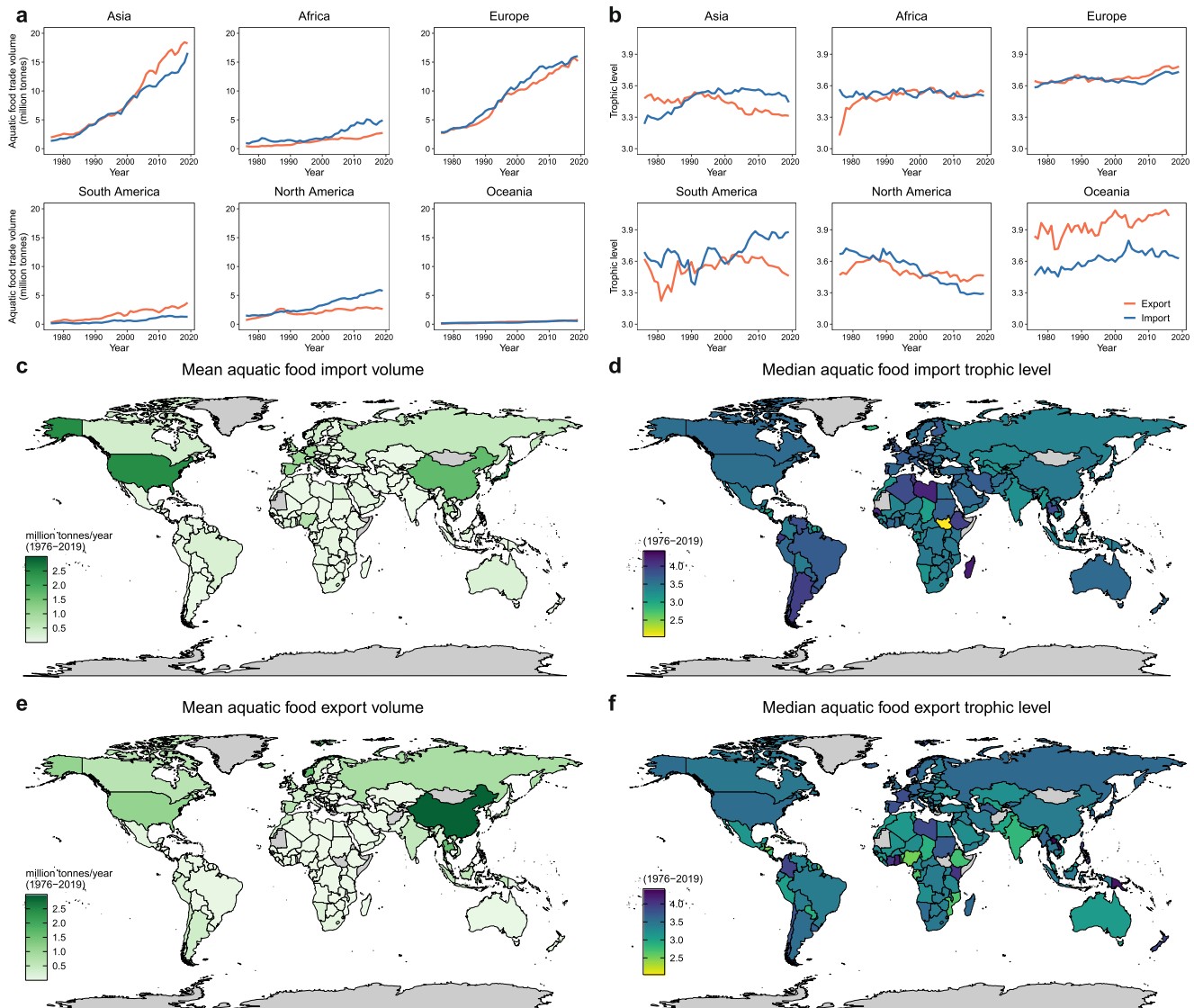

**Fig. 2 | Global aquatic food trade volume and trophic level from 1976 to 2019.**
**a** Trends of aquatic food import and export volume in different continents.
**b** Trends of aquatic food import and export trophic level in different continents.

The mean country-level aquatic food import volume (**c**) and export volume (**e**). The median country-level aquatic food import trophic level (**d**) and export trophic level (**f**). Countries in gray: No data available.

intensive farming[40]. Asia, and China in particular, play a crucial role not only because of its significant contribution to global aquaculture production, but also because it accounts for the world's largest quantity of farmed low-trophic level species (Supplementary Figs 1–4 and 5b). Nonetheless, it should be noted that aquatic food consumption in this study does not refer to the quantity effectively eaten but to the theoretical maximum live weight available before consumption.

Regional consumption and HATL trends show large variation (Fig. 1c, d). In line with global trends, Asia's per capita consumption of aquatic foods has increased rapidly, mainly driven by China, whereas the HATL declined at an approximate rate of 0.08 per decade; approximately 1.4 times the global rate (Fig. 1a, c, d and Supplementary Fig. 4c, d). By contrast, the per capita consumption of aquatic foods in Europe and South America rose rapidly for a short time and then fell sharply from the 1990s, coinciding with the increase in their HATLs. North America experienced a slight overall increase in per capita consumption of aquatic foods but a decrease in HATL. Interestingly, Oceania was the only region where both per capita consumption of aquatic foods and HATL increased over the study period, although recent years suggest a decline in per capita consumption (Fig. 1c, d).

Currently, Asia has the lowest HATL, Africa has the lowest per capita consumption of aquatic foods and a low HATL, and Europe and Oceania are the regions with the highest per capita consumption of aquatic foods and HATL (Fig. 1c–f). In general, we find regions with more developing countries to have lower per capita consumption and lower trophic level of aquatic foods than regions with more developed countries. Nonetheless, trade seems to mediate these apparent strong imbalances in aquatic food consumption across regions and countries (see next section).

**Aquatic food trade**
In the past decades, international trade in aquatic foods has risen across all continents, especially in Asia and Europe, which represent the two major trading regions (Fig. 2a). Historically, aquatic food trade has been dominated by a few countries, such as China, USA, Norway, Thailand, and Japan (Fig. 2c, e and Supplementary Table 1). Since the World Trade Organization (WTO) was founded in 1995, Asia and South America have been the major trade surplus regions (i.e., higher exports than imports), whereas Africa and North America have been the major trade deficit regions (i.e., higher imports than exports) (Fig. 2a). The

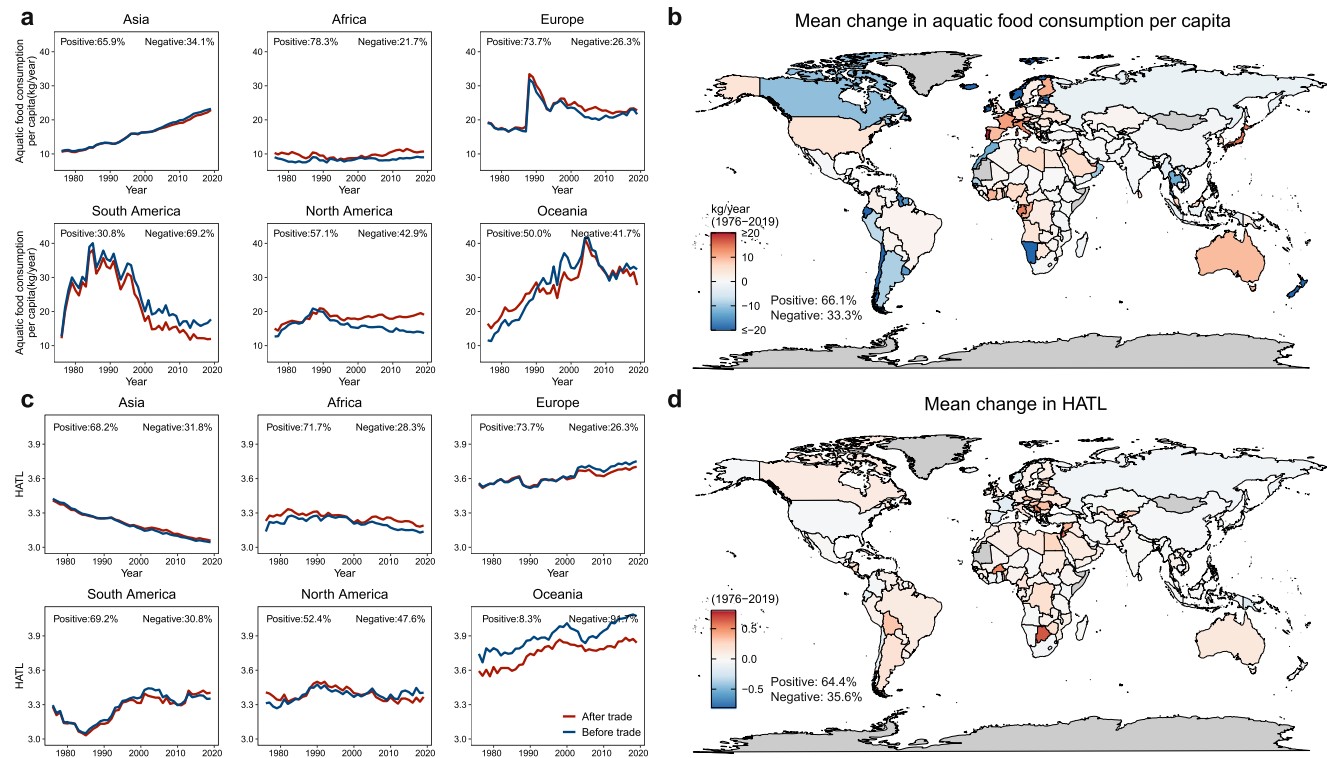

**Fig. 3 | Impact of trade on aquatic food consumption per capita and HATL.**
**a** Trends of continental per capita consumption of aquatic foods before and after trade. **b** The mean country-level change in annual per capita consumption of aquatic foods after trade from 1976 to 2019. **c** Trends of continental HATL before and after trade. **d** The mean country-level change in annual HATL after trade from 1976 to 2019. All country-level changes in per capita consumption of aquatic foods and HATL are the post-trade value minus the pre-trade value year by year.

Percentage values indicate the proportion of countries affected by trade (positively or negatively) in each region (**a**, **c**) and globally (**b**, **d**) (for details, see Supplementary Table 2). 'Before trade' represents the maximum available live weight per capita of domestically produced aquatic foods, while 'after trade' refers to the apparent consumption patterns after completing trade transactions. HATL human aquatic food trophic level. Countries in gray: No data available.

share of imports in total aquatic food consumption has been rising in developed countries, which have good supply chain infrastructures and more consumers who can afford to buy imported high-value species[2,14]. Developing countries are becoming increasingly prominent in the supply of aquatic products and becoming increasingly important as supply chain intermediaries, importing raw materials and re-exporting processed or value-added products[14]. For example, although China is also one of the largest importers and exporters (in terms of live weight), more than two-thirds of these imports are raw materials that are processed and re-exported[23].

The trophic level of continental trade from 1976 to 2019 was generally above 3.3 across all continents (Fig. 2b), while the country-level median import and export trophic level was above 3 for most countries (Fig. 2d, f). These figures suggest that most of the exported aquaculture and capture fisheries products consist of high-trophic level species for international markets, particularly from Europe and Oceania. Interestingly, Asia and South America's aquatic food import trophic levels have surpassed those of exports in recent decades, whereas North America and Europe show the opposite trend (Fig. 2b). Meanwhile, although the trophic level of imports and exports has remained similar in Africa, its import volume is gradually increasing faster than the export volume (Fig. 2a, b). Together, these results indicate that the trade structure and consumption features of aquatic foods in these regions are changing. First, a more robust demand for aquatic species with higher trophic level (i.e., higher value) is apparent in Asia and South America as production and incomes rise, gradually redirecting products once produced mainly for exports toward domestic markets[15,38,41,42]. Second, the trend in developing countries to export high-value aquatic foods in exchange for low-value aquatic

foods from industrial fisheries is being reversed[21]. While significant quantities of high-trophic level species (e.g., salmonids) are traded and continue to grow, the trade volume of low-trophic level species (e.g., tilapia and shrimp) has also increased drastically[14], which has helped keep the global import trophic level largely stable between 3.5 to 3.6 (Supplementary Fig. 6). Asian and South American countries, especially in East Asia, have been the central supply regions for relatively low-trophic level species in recent decades (Fig. 2a, b, d, f).

## Trade impacts on aquatic food consumption patterns
Differences in per capita consumption of aquatic foods and HATL by country between 'before trade' (i.e., consumption stage before trade transactions) and 'after trade' (i.e., apparent consumption patterns after completing trade transactions) reveal the rapidly increasing volume and shifting trade features in various regions that have affected both per capita consumption of aquatic foods and HATL in most parts of the world. Continentally, the per capita consumption of aquatic foods has decreased in Asia and especially South America after trade (Fig. 3a). Meanwhile, the HATL of these two regions remained nearly unchanged (Fig. 3c). After 2000, Asia and South America were two major trade surplus regions (i.e., after-trade aquatic food consumption was lower than before trade), supplying increasing quantities of low-trophic level species and playing an essential role in boosting per capita consumption of aquatic foods in the rest of the world (Figs. 2a, b, and 3a, b). North America has maintained its per capita consumption of aquatic foods over time, despite producing less, through a gradually increased reliance on imports (Figs. 2a and 3a). For example, the import share of total aquatic food consumption in the USA rose from one-third in 1961 to nearly three-quarters in 2019[14].

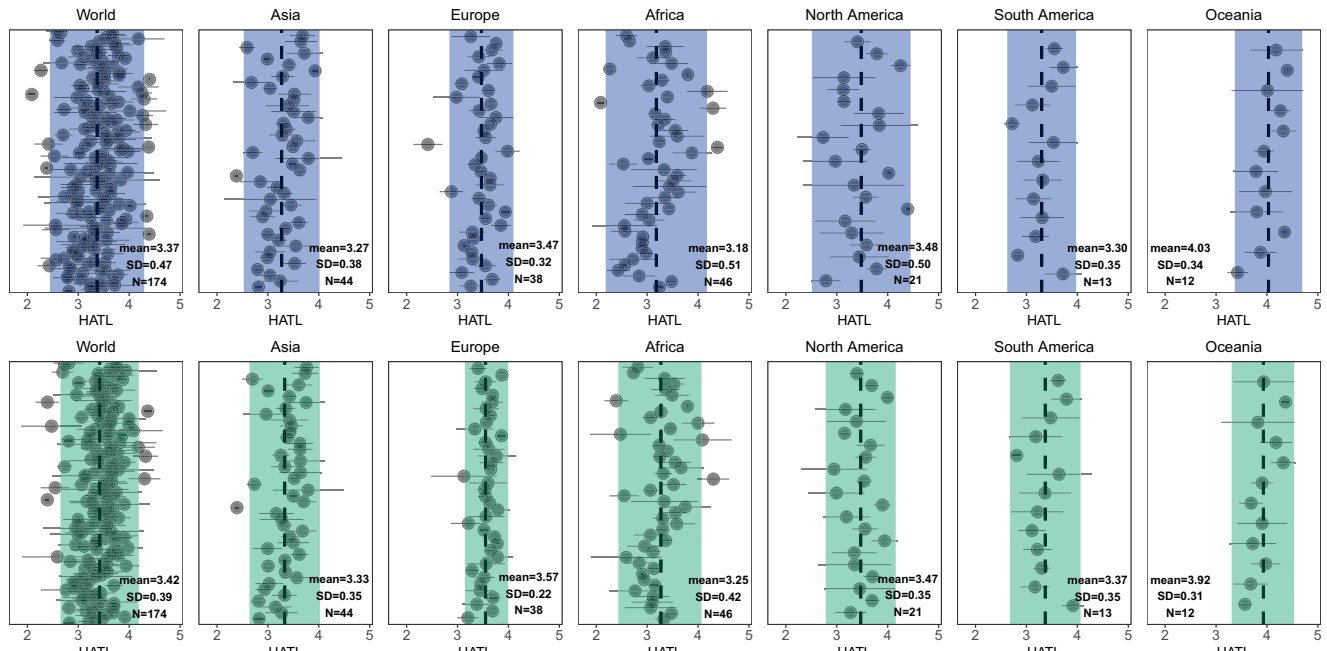

**Fig. 4 | Global and regional HATL dispersion before and after trade from 1976 to 2019.** In all plots, each point shows the mean estimate, and error bar shows 95% reference range (mean ± 1.96 SD) for each country. The shaded blue column (before trade) and green column (after trade) indicate the 95% reference range (mean ± 1.96 SD) for all countries in different continents. The dotted lines correspond to the HATL averages of all countries in different continents. Numbers refer to the number of countries in each region (N = 174 total). 'Before trade' represents the maximum available live weight per capita of domestically produced aquatic foods, while 'after trade' refers to the apparent consumption patterns after completing trade transactions. HATL human aquatic food trophic level.

Conversely, import and export volumes in Europe have grown at similar rates, resulting in only slight differences in per capita consumption of aquatic foods before and after trade in recent years. Countries in both North America and Europe, mostly developed, have experienced mild declines in post-trade HATL over the past decade (Fig. 3c). Meanwhile, although differences in per capita consumption of aquatic foods before and after trade in Oceania are small, Oceania's HATL has decreased considerably after trade (Fig. 3a, c), because the trophic level of exports is significantly higher than that of imports (Fig. 2b). Finally, the fact that Africa is notably the only region where both post-trade per capita consumption of aquatic foods and HATL have increased underscores the importance of trade in improving aquatic food availability in Africa.

At the national scale, although Asia is a trade surplus region, more than 60% of Asian countries have benefited from trade in their aquatic food consumption (Fig. 3a, c). Except for South America, per capita consumption of aquatic foods increased in most countries of all continents (Fig. 3a). Similarly, trade has also increased HATLs in most countries across continents except Oceania (Fig. 3c). In particular, over 70% of countries in Africa and Europe benefit from trade in both aspects of aquatic food consumption. Indeed, the dominance of certain countries in international fish trade masks the importance of trade for Africa, where aquatic food demand has grown faster than supply, resulting in an increase in the import share of consumption from 16% in 1970 to 39% in 2017 as production from domestic fish capture has either stagnated or been exported[18,43]. Globally, international trade has increased the availability and trophic level of aquatic foods in most (>60%) countries over the past decades (Fig. 3b, d). Furthermore, the heterogeneity in the HATL of countries declines globally after trade, especially in Europe, North America, and Africa, while the mean HATL has increased in most continents (Fig. 4). Overall, our findings suggest that international trade has both reduced geographic differences in HATL and improved the aquatic food consumption in most parts of the world. Trade will therefore be an important part of a transition to sustainable fisheries.

## Discussion

The need to increase aquatic food diversity and supply to help achieve global food and nutrition security goals within environmental boundaries is a global consensus. In recent decades, the rapid growth in aquatic food globalization and consumption has been driven by increased trade liberalization and facilitated by advances in food processing and transportation technologies[2,14]. Aquatic foods are among the most highly traded commodities[4,5], comprising nearly 10% of all food trade by value[21]. Obviously, international trade in fishery products is playing a significant role in shaping global aquatic food harvesting and consumption[4]. However, aquatic foods have often been excluded from previous studies on detailed global food trade due to the difficulty in reconciling species-level production and trade data[18,22].

By developing a species-level mass balance dataset and a trophic level identification dataset for 174 countries, we first reveal precise country-level dynamics in aquatic food consumption patterns, identify the features of aquatic food trade, and quantify the effect of trade on aquatic food consumption globally. From 1976 to 2019, global per capita consumption of aquatic foods has increased significantly, but the HATL is declining (from 3.42 to 3.18). The rapid development of aquaculture and its significantly lower average trophic level relative to capture fisheries primarily contributed to this trend (Fig. 1a, b). Meanwhile, international trade has played an important role in harmonizing the global consumption of aquatic products, increasing the availability and trophic level of aquatic foods in most countries (especially for Africa and Europe), and reducing HATL heterogeneity worldwide. In this study, we did not consider mollusks and aquatic plants. Given that these groups have a lower trophic level than most other aquatic foods and considering the recent increase in the consumption of mollusks and aquatic plants[2,15,29], our estimate of the current consumption trophic level can be considered conservative (i.e., the inclusion of these groups will likely lower our estimated HATL). Nonetheless, the continuous decrease in the effective trophic level of the majority of farmed species[29] coupled with the increasing proportion of low trophic level species in the diet are encouraging for

progressing towards reducing dependence on multiple marine ingredients (e.g., fishmeal and oil). Further improvements in resource conversion efficiency on this basis will yield even greater results. Furthermore, aquatic foods not only provide comparatively higher nutrient richness across multiple micronutrients, vitamins, and long-chain polyunsaturated fatty acids relative to terrestrial animal-source foods[6], but they also typically have lower environmental footprint compared to other animal-sourced foods[44,45]. The observed trends of increasing global contributions from aquatic foods suggest a promising future for more sustainable global diets.

The impacts of the aquatic food trade on food security and well-being continue to be a subject of intense debate[3,20]. While some claimed that fish trade is beneficial for marginal and vulnerable local communities, others denounced a negative impact of fish exports on food security for these communities[20,46]. Our study supports the conclusion that global trade can improve food availability by allowing most countries to access larger quantities and higher trophic level aquatic foods that otherwise are domestically unavailable. However, the direct contribution of trade to the food system in vulnerable population groups is limited because the beneficiaries tend to be high-income groups as most exported products consist of high-trophic level species (high-value species) for international markets[47]. Furthermore, differences in dietary habits, income levels, natural resource conditions, and other aspects among different regions can lead to variation in the cost of aquatic products and consumers' affordability. These situations highlight the need for fair, transparent, sustainable, and adaptive trade and market policies to ensure that more segments of society benefit from international trade[46].

Until now, the problem of imbalances in the growth of demand and aquatic food supply remains prevalent across regions, countries, and income groups[18,48]. Many people remain under multiple forms of malnutrition and per capita consumption of aquatic foods is far below the world average[6,14]. Geography plays a major role in explaining these differences[14]. As the world may be approaching the constraints of a finite, global, aquatic food production capacity[49,50], sourcing trajectories from all countries must be considered together[38]. Based on our findings, it is evident that trade has played an important role in harmonizing the global aquatic food consumption. Nevertheless, there are still some countries where per capita consumption of aquatic foods has not improved after trade and remains quite low (Figs. 1e and 3b). Heightened attention and concerted efforts for context-specific mitigation should therefore be given in the future to these countries. To this end, attaining globally equitable trade distribution patterns as well as more harmonized trade environment and policies should be a priority. Despite the important progress attained in research on aquatic food production and trade, significant challenges persist in achieving a comprehensive understanding of the outcomes of aquatic food trade[5,17–21]. Our work adds another piece to this puzzle by identifying the implications of trade for contemporary changes in global aquatic food consumption patterns, highlighting the increased availability and trophic level of consumed aquatic foods in a majority of countries with reduced differences in HATL despite the important remaining inequalities. These results provide an important foundation to guide future research on the globalization of aquatic food systems and the impacts of trade on food security.

## Methods

### Preprocessing of fisheries datasets

Three original global fisheries statistic datasets (aquaculture, capture, and trade) were taken from the FishStatJ software[51]. The original trade dataset contained more than 100,000 commodities, each mixed by species/species group, preservation, and preprocessing method, such as 'Catfish fillets, frozen'. The original aquaculture and capture datasets also had more than 25,000 items. The earliest coincident year of these three datasets was 1976. We removed all items of negligible importance (i.e., total volume <100t) from 1976 to 2019 in all original datasets to facilitate the definition of the live weight conversion factor for trade commodities and the trophic level of species or species groups in aquaculture and capture datasets according to the fishing area in each country (see next two sections). Removed items accounted for a total of 0.001% of aquaculture production, 0.004% of capture production, and 0.044% of trade volume.

The term 'aquatic foods' is used throughout this study to denote all freshwater and marine fish, cephalopods, and crustaceans. Brackish fish were identified as freshwater fish or marine fish according to the major fishing area in the aquaculture and capture datasets. Algae, aquatic plants, mollusks (Bivalvia, Gastropod, Barnacle, and Ascidiacea), echinoderm, cnidaria, miscellaneous aquatic animals (such as turtles, frogs, and mammals), and reported inedible species were not considered for this study. Although mollusks and algae accounted for a significant proportion of fisheries output in live weight, especially in aquaculture, they comprised a very small proportion in edible weight[15,18,51,52]. Further, we were unable to find publicly available preservation factors (i.e., the ratio of edible portions to final product live weight) for the different processing methods of these groups.

In the trade dataset, we deleted commodities for which no live weight conversion factors were available, such as fish sausage and fish cake. We have limited our focus to edible aquatic foods and have discarded commodities such as fishmeal and fish oil that are unsuitable for direct human consumption. Therefore, to mitigate the impact of specific fish used in the production of fishmeal and fish oil on aquatic food consumption in several countries, we removed the world's most important forage fish (i.e., *Engraulis ringens*) from the capture dataset. *Engraulis ringens* had comparatively a massive production but is rarely used for direct human consumption in Chile and Peru[53]. Additionally, due to the absence of accurate or even approximate long-term time series data on *Engraulis ringens* consumption proportions, accurate country-level estimations of the theoretical maximum live weight availability for this species were not possible. While many other small pelagic fish (e.g., herrings and sardines) are also partly reduced to fishmeal and fish oil, their volumes were relatively low and some were used directly for human consumption. Therefore, we retained these fish species for analysis. Following these data processing steps, 174 countries with available aquaculture, capture, and trade data accounting for 95.8% of the world's fisheries and aquaculture production from 1976 to 2019 were included for subsequent analyses.

To determine whether the removal of production and trade items with a total volume <100t from 1976 to 2019 would result in highly unequal impacts on small and large countries, we examined the proportion of removed items in production and trade for all 174 countries. We discovered that in 97% of countries, the ratio of excluded items to total production is below 1%. Additionally, in 95% of countries, the proportion of excluded items to total trade volume is less than 3%. Therefore, the removal of items with a total volume <100t would not have significant unequal impact on different countries.

### Live weight conversion of traded commodities

Trade data were reported at the species-level and broader commodity groups (e.g., 'Salmons nei, Eels nei, and Tunas nei') and labeled based on processing (e.g., fresh, frozen, dried, fillets, etc.). In order to back-transform processed product weight into whole-animal live weight equivalents, we first extracted commodity species or species groups, preprocessing and preservation methods from original commodities in trade dataset. We then used conversion factors from ref. 54. Hortle[55], expressed as live weight (kg) of aquatic animals required to make 1 kg of product, calculated as the product of the preprocessing factor and the preservation factor. The preprocessing factor represents the ratio of live weight to edible portions after cleaning (beheading, gutting, etc.). The preservation factor is defined as the ratio of edible portions to final product weight. Aquatic products for which no preservation

and/or preprocessing methods were reported were assumed to refer to fresh units and/or whole-animal. All conversion factors are listed in Supplementary Table 4. After conversion, the average live weight ratio of the total imports to the total exports was $0.99 \pm 0.04{:}1$ over the study period (1976–2019), suggesting that the conversion factors were relatively reliable, because in theory the world's total import volume should equal total export volume. All processing details and results were available in Supplementary Data 1.

### Trophic level identification

Items in the aquaculture and capture production datasets were often not provided at the species-level but were more generically referred to by a species-group name (e.g., 'Groupers nei'). In such cases, we searched each species group in Fishbase (www.fishbase.org)[56] using the 'Common name is' function to screen out all species identified by that common name and with clear economic value according to the reported main fishing area by country. In a few cases, no species were found. In those rare cases, we used the 'Common name ends with' function instead to identify the species involved. The trophic level of each species group was then calculated as the mean trophic level of all identified species for that group (Supplementary Data 2 and 3). The trophic level of specific species was extracted directly from Fishbase.

For unidentified 'Freshwater fish nei' and 'Marine fish nei' items in production dataset, the trophic level was obtained by the production-weighted average of trophic levels of identified fish species or species groups according to the fishing area in each country. For those few countries that did not have any identified species or species groups, we weighted and averaged the trophic level of identified 'Freshwater fish nei' and 'Marine fish nei' items from all countries in the same fishing area. The trophic levels of marine 'Pelagic fish nei' and 'Demersal fish nei' were considered the same as 'Marine fish nei'. The trophic level of unidentified items was based on the trophic level and total production of identified species or species groups:

$$TL = \frac{\sum TL_n * W_n}{\sum W_n} \qquad (1)$$

where $TL$ is the trophic level of unidentified items, $TL_n$ is the trophic level of involved species or species group to be weighted, and $W_n$ is the total production of involved species or species group from 1976 to 2019.

In the import dataset, the trophic level of specific species was also extracted directly from Fishbase. For generic import commodities, such as 'Tunas nei', the trophic level was calculated as the weighted average of trophic levels based on global production data for all involved species in the group. It was assumed that the most productive species were the most likely to enter the trade flow. When several trade commodities included more than three species or species groups (e.g., 'Herring, anchovy, sardine, sardinella, brisling/sprat, mackerel, Indian mackerel, seerfish, jack & horse mackerel, jack, crevalle, cobia, silver pomfret, pacific.saury, scad, capelin, etc.'), we kept the first three species or species groups (i.e., 'Herring, anchovy, and sardine') based on the importance ranking assumption. The trophic level of this commodity type was the weighted average of trophic levels of all involved species in these three groups. Moreover, the trophic level of import 'marine fish nei' and 'freshwater fish nei' commodities were obtained as the weighted average of trophic levels of all 'marine fish nei' and 'freshwater fish nei' items in the global production dataset. Since we subtracted exports and reexports from production and imports after identifying the trophic level of items in production and import datasets, the trophic level of exported commodities could be acquired when the species-level mass balance was finished (see 'Species-level mass balance from FAO statistics' below).

In the production and trade datasets, 'Crustacean' was given a trophic level of 2.5, 'Cephalopod' of 3.0, and 'Demersal percomorphs nei' of 4.0. Although the trophic level of cultured species is related to

the feed composition and diverges in effective trophic level from their wild counterparts[29,31], we did not consider this due to the lack of sufficient data. Similarly, the trophic level of the same wild capture species was considered constant across different seas and time periods.

### Human aquatic food trophic level

In this study, human aquatic food trophic level (HATL) was considered a composite metric that reflects human aquatic food diet patterns simply and synthetically. We calculated the HATL using trophic level and live weight data of consumed species or species groups[57,58]:

$$HATL = \frac{\sum TL_i * W_{ij}}{\sum W_{ij}} \qquad (2)$$

where $TL_i$ is the trophic level of species or species group $i$, and $W_{ij}$ is the live weight of species or species group $i$ in year $j$. HATL is the quantity-weighted average of trophic levels of species or species groups consumed in a particular year by country. Similarly, the trophic level of aquaculture, capture, imports, and exports were calculated using Eq. (2) as defined above.

### Species-level mass balance from FAO statistics

We subtracted export weights from production in four sequential steps and present the details of species-level mass balance from FAO statistics in Supplementary Fig. 7. Once the exports and reexports were subtracted from production and imports separately, the remaining weight was assumed to represent apparent consumption per commodity group in each country.

**Mass balance principle 1.** Each reexported commodity was matched one-to-one with the imported commodity with the same common name. All unmatched commodities were matched with each country's 'Fish nei' or 'Freshwater fish nei' or 'Marine fish nei' items. Step 1 produced the remaining imports data.

**Mass balance principle 2.** We combined aquaculture, capture, and remaining imports data (produced in step 1) into one dataset. The same species or species groups in each country were summed, and the trophic levels were simultaneously weighted and averaged.

Each exported commodity was matched one-to-one with the combined data items with the same common name (i.e., aquaculture + capture + remaining imports). All unmatched commodities were matched with each country's 'Fish nei' or 'Freshwater fish nei' or 'Marine fish nei' items. This step was deemed necessary, because we believed that global reexport volumes were underestimated. For example, Asche, et al[23]. estimated that 74.9% of China's seafood imports were reexported, but there were very few records of China's reexports in the FAO trade dataset, so some reexports in the trade dataset must be only roughly marked for exports or omitted. Thus, we combined aquaculture, capture, and remaining imports data to subtract exports. Step 2 produced the remaining exports 1 (negative value) and consumption volume 1 (positive value).

**Mass balance principle 3.** Species-level remaining exports 1 commodity was matched with the generic item (e.g., 'Yellow tuna' can be matched with 'Tuna nei), and generic commodity was matched with all contained species (e.g., 'Tunas nei' could be matched with all tuna species) in consumption volume 1. Some detailed matching information between several specific exported commodity groups and production items can be searched in Supplementary Table 3.

All unmatched remaining exports 1 commodities were matched with the 'Fish nei' or 'Freshwater fish nei' or 'Marine fish nei' items in each country. A few remaining unmatched 'Fish nei' or 'Freshwater fish nei' or 'Marine fish nei' were matched with the three most productive items in each country. Each remaining exports 1 commodity was deducted from

consumption volume 1 according to the proportion of the output of all matching items each year. Step 3 produced the remaining exports 2 (negative value) and consumption volume 2 (positive value).

**Mass balance principle 4.** Except for each country's 'Freshwater fish nei', 'Marine fish nei', 'Fish nei', 'Pelagic fish nei', and 'Demersal fish nei' were matched with the three most productive items, all remaining export 2 commodities were matched with the 'Freshwater fish nei' or 'Marine fish nei' or 'Fish nei' items in consumption volume 2. Each remaining exports 2 commodities was deducted from consumption volume 2 according to the proportion of the output of all matching items every year. Step 4 produced the remaining unexplained exports (negative value) and final consumption volume (positive value).

Theoretical reexported commodities and volumes can be obtained by comparing the original imports with the subtracted imports. Likewise, the theoretical exports are the difference between the final consumption volume and the original data (i.e., aquaculture + capture + remaining imports).

### Mass balance rationality analysis

Theoretical exports plus theoretical reexports were hereafter called exports. After the process described above, the total exports accounted for $95.3 \pm 0.9\%$ (mean ± SD) of the total original exports (including exports and reexports), and the ratio to imports was $0.96 \pm 0.03$:1 from 1976 to 2019 (Supplementary Fig. 6 a). We also calculated the proportion of export volume to original export volume for all countries and regions in each year, with a minimum average of 72.6% from 1979 to 2019, and over 90% of countries and regions exceeding 85%. The remaining unmatched exports might be due to imprecise conversion factors, imperfect matching, and reporting errors[23], which only account for $1.1 \pm 0.4\%$ of total consumption from 1976 to 2019 (Supplementary Fig. 6 b). Due to these inevitable errors mentioned above, it is almost impossible for global import and export trophic levels to be theoretically identical. Similarly, Asche, et al.[23], Kroetz, et al.[22] also indicated that it was difficult to reconcile aquatic food production and trade data because of mismatches in species- versus product-level reporting and weight losses during processing. In this study, the average difference between trophic levels of imports and exports is only 0.02 (Supplementary Fig. 6c), and global human aquatic food trophic level trends before and after trade almost coincided (Supplementary Fig. 6d). Thus, theoretical exports could be a good proxy for original exports, and remaining unmatched exports would not affect global human aquatic food consumption patterns.

### Limitations and uncertainties

Although we provided more detailed information on aquatic food production, trade, and consumption than the FAO Food Balance Sheets, which contain live weight of broad taxonomic groups, we acknowledge that our study has some limitations due to a lack of sufficient data. First, it is inevitable that production and trade data in some of the countries analyzed are inconsistent or not perfectly processed, leading to over- or under-estimated apparent consumption and HATL. Second, to see the magnitude of the differences between both estimates, we compared our balanced average annual per capita consumption to the average annual per capita supply in the FAO Food Balance Sheets. A total of 66.7% of countries had an average difference magnitude below 20%, and a large majority (73.6%) fell below 30%. These differences can mainly be attributed to the fact that the FAO accounts for non-food uses and variations in stocks, as well as differences in live weight conversion factors. Although we removed the world's most important forage fish (i.e., *Engraulis ringens*) from the capture dataset, we couldn't remove the effect of all other existing fish species used in the production of fishmeal, fish oil, and other non-food uses, as well as variations in stocks on aquatic food consumption in all countries. Third, given the increasing rates of food loss and waste

linked to the increasing industrialization and income of countries, it is likely that the effects of trade may have been offset to some extent by supply chain leakages, resulting in some cases in smaller gains than those estimated here or perhaps even reduction in actual consumptions. Therefore, our trade analysis provides a baseline for potential food security gains, but we recognize that other factors not addressed here are also crucial to draw a strong conclusion on how trade affects food security and should be subject of future studies. Fourth, we didn't account for the difference between the wild trophic level of farmed species versus actual trophic level based on what they were actually being fed[29]. These patterns were subject to both strong temporal trends and significant spatial variation. Finally, live weight conversion factors could vary over time and geographies due to differences in processing technologies, even for the same species.

### Reporting summary

Further information on research design is available in the Nature Portfolio Reporting Summary linked to this article.

## Data availability

The raw data used in this study are publicly available (see Methods). The processed data generated in this study are either included in the Supplementary Information or available in the figshare repository (https://doi.org/10.6084/m9.figshare.21692186.v3).

## Code availability

The R codes used to conduct the study are available in the GitHub (https://github.com/zhaokangshun/Effect-of-trade-on-global-aquatic-food-consumption-patterns.git) and archived through Zenodo (https://doi.org/10.5281/zenodo.10129727).

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

## Acknowledgements

This research was supported by the National Key R&D Program of China (grant no. 2018YFD0900904). K.Z. was funded by the China Scholarship Council. J.X. acknowledges the support received from the International Cooperation Project of the Chinese Academy of Sciences (grant no. 152342KYSB20190025) and the National Natural Science Foundations of China (grant no. 31872687).

## Author contributions

K.Z. and J.X. conceived the idea. S.D.G. contributed to the study design. K.Z., M.Z., and J.X. contributed to the acquisition and analysis of data.

K.Z., S.D.G., and J.G.M. contributed to the interpretation of results. K.Z., S.D.G., M.Z., and J.G.M. wrote and edited the manuscript.

## Competing interests

The authors declare no competing interests.
