## [Peer Review File · Nature Communications]

Reviewers' Comments:

Reviewer #1:

Remarks to the Author:

The authors cover an important topic that is worthy of publication. However, in my opinion there are some errors that need correcting.

1. The title of the paper is misleading (and is the opinion of the authors and not of most NGOs) and I would suggest it be changed to "Effect of trade on global aquatic food consumption patterns";
2. The paper excludes shelled mollusks (bivalves, clams, mussels, oysters etc) and aquatic plants that are also consumed as food - so really it only covers selected aquatic animal foods;
3. It is a pity that the authors did not consult with the FAO Food Balance Sheets (latest data for 2020) in their analysis as this includes all aquatic food products on a live weight basis, including all mollusks and aquatic plants - these food products would bring down the trophic level of global aquaculture production even further;
4. In view of the above, the paper is not a review of global aquatic food consumption patterns, but of selected food products - this should be clearly stated;
5. Statements implying that higher trophic level aquatic species have a better protein quality are not true and should be stated as such (line 67, 74, 149, 152 etc) - the implication being that lower trophic level species have a lower nutritional quality - in fact the reverse is probably true as wild-caught high trophic level species are more likely to be contaminated with environmental contaminants such as POPs, heavy metals, and microplastics;
6. The authors investigate the contribution of aquatic animal food products to total animal protein supply;
7. Trade impacts on aquatic food consumption patterns - no mention is given to the costs of aquatic food products and their affordability or not to consumers - in most South American countries poultry is the cheapest source of animal protein whereas in Asia it is the reverse - dried fish usually being the cheapest source of animal protein to the rural poor, including small captured pelagic fish in West African countries;
8. Finally, the exclusion of the Peruvian anchovy from the analysis (Methods - 263, 289 etc) is not recommended - an increasing proportion being directly consumed as food (it is a food grade product and as such not labeled as forage fish in my opinion), and also have a low trophic level which would influence final trophic level calculations;

Reviewer #2:

Remarks to the Author:

This paper connects production with consumption of aquatic foods and explores how trade influenced HATL and consumption patterns. I found the analysis and paper interesting and well written, further increasing our understanding of how trade within the aquatic food system contribute to alleviating nutritional insecurity. As such, it is an important contribution to the scientific community and policymakers. Further, I believe this paper should be accepted for publication, pending some major comments I provide below. These are concerned mostly with the interpretation of the results and conclusions.

The authors used changes in consumption per capita and trophic levels to infer on changes to food

security. However, such an inference is only partial. For one, fish's nutritional contribution is more a product of its micronutrient contribution (concentrations) than its protein. The variation in protein concentration among fish (teleost) is very small, but as the authors rightly mentioned at the beginning there are small fish (lower trophic level) with phenomenal micronutrient concentrations. So, I would suggest amending the text in the cases where high trophic level is used synonymously with "higher quality". The importance of tracking trophic levels is well noted but it's difficult to infer on the nutritional quality as a whole just by looking at trophic levels.

The authors note positive contributions of trade to consumption per capita amongst most of the population in the world, but looking closely reveals very trivial contributions (Fig 3) for most continents (except North and South America). If the authors would have highlighted the changes to consumption of the 'before trade' and the 'after trade' in percentages, I would assume we would see very small values in many cases, maybe within the range of uncertainties (which are not noted here). The authors note that >60% of countries experienced an increase in food availability, but how much of those are above, say, 20% change or any other uncertainty level? It is important to note that food availability is only one dimension of food security and other important factors, especially within countries, such as access, affect food security too. Moreover, the authors track food availability, not true consumption at household level. Given the increasing rates of food loss and waste as the industrialization and income of countries increase some of those food gains they noted may be offsetted by supply chain leakages, resulting in smaller gains or perhaps even reduction in actual consumptions. Therefore, the trade analysis offers a glimpse to possible food security gains, but other factors, not addressed here, are as crucial in order to draw a strong conclusion on how trade effects food security. The authors must acknowledge this in their conclusions and limitations more explicitly (not only in the Methods). Perhaps even changing the title of the paper.

Lastly, the authors highlight the aquatic food production and consumption amongst nations, but maybe adding another figure that notes the supply chain flows – i.e., connection of aquatic foods between specific countries (circular flow chart) – can also be revealing in this analysis.

General comments

Line 129 might be necessary to define what trade deficient/surplus means

Fig 2a and b – please make sure the colors of export and import are similar on both subplots

Fig 2d,f I would use the same color code as in fig 1 for HATL for consistency and easiness of comparison.

Line 168: Is "before trade" actually the domestic consumption per capita of what is produced locally? Maybe change the wording to make it clearer.

Line 169 maybe add a sentence that says that you've explored the differences between "before" and "after trade" on countries' consumption per capita and HATL to identify patterns of changes due to trade. It's not clear enough.

Line 170 unnecessary comma after especially.

Line 172: please write "after/before trade" as a phrase using parenthesis, otherwise its confusing.

Line 235-237 This is inaccurate. Energy cascade among trophic levels of 10% is generally correct, but GHG of wild capture fisheries is dictated by fishing efforts and when it comes to aquaculture with specific management practices, feeding practices (prices, access) and regions, there are exceptions to trophic level being a predictor for environmental impacts. I would cite the Gephart et al, Nature 2021 for the environmental impacts of aquatic foods.

Line 242-243 only in the past? It's still an on-going question in the present. This present research adds another piece to the puzzle, but it certainly does not resolve this. Trade impacts food security on multiple dimensions, not all of them addressed here.

Line 252 imbalances of what? Not clear

Reviewer #3:

Remarks to the Author:

This paper uses seafood production and trade data from the FAO statistical database and FishBase to develop species-level mass balance data and a trophic level dataset. The authors use the datasets to calculate per capita aquatic food consumption and an aquatic food trophic level at the country level. The authors examine how aquatic food consumption and the aquatic food trophic level is impacted by trade of aquatic food. The work is important given global levels of traded aquatic food and the mixed literature with regards to the impacts of trade on aquatic food security. The results show that the mean aquatic food trophic level has declined. The results also show that international trade has increased aquatic food consumption and the trophic level of most countries as well as reduced geographic differences in trophic levels.

The paper is well written, the methodology is sound and described in great detail, and I expect the paper will be of high interest to those in and out of the field. My main concern is related the authors' conclusion that a decline in aquatic trophic level "is potentially good news from the perspective of the environmental footprint of food production" (line 232-233). The relationship between trophic level and environmental performance is much more nuanced. Gephart et al. (2021, Nature) show that several low trophic level species, like tilapia and shrimp, have higher environmental impacts in many dimensions compared to higher trophic level species like salmon. More so, the environmental performance of aquacultured species is intricately linked to feed inputs and conversion efficiencies which doesn't necessarily correspond to trophic level (Cottrell et al. 2021, Reviews in Aquaculture). I would like to see the authors interpret their results in light of this.

The authors should also compare the per capita aquatic food consumption estimates in this paper with the FAO food balances and discuss thoughts on why their estimates are in some cases very different from the food balances (e.g., the authors' estimates of per capita consumption for China and Indonesia are much lower).

REVIEWER COMMENTS

NOTE: We address each comment individually below with the original comment in black text and our response in blue text. To help locating our changes in the manuscript, we indicate the lines with “L”.

Reviewer #1:

The authors cover an important topic that is worthy of publication. However, in my opinion there are some errors that need correcting.

We are grateful for your thorough review and your constructive feedback, which have helped improving our manuscript. We hope to have addressed all your concerns and that you find our revision satisfactory.

1. The title of the paper is misleading (and is the opinion of the authors and not of most NGOs) and I would suggest it be changed to "Effect of trade on global aquatic food consumption patterns";

Agreed. We have changed the title to "Effect of trade on global aquatic food consumption patterns" as suggested.

2. The paper excludes shelled mollusks (bivalves, clams, mussels, oysters etc) and aquatic plants that are also consumed as food - so really it only covers selected aquatic animal foods;

Thank you for your comment. Our primary consideration for not including these groups is that the edible proportion in live weight of shelled mollusks and algae is very low, and there is a significant variation among different mollusk species. Moreover, they comprised a very small proportion in edible weight of total aquatic food production and consumption¹⁻³. For instance, on an edible-weight basis, mollusks and algae comprised only 6% and 7.6%, respectively, of total aquaculture output^{1,4}. We have added some text to the corresponding section of the Methods to clarify this point (L301-308).

“Algae, aquatic plants, mollusks (Bivalvia, Gastropod, Barnacle, and Ascidiacea), echinoderm, cnidaria, miscellaneous aquatic animals (such as turtles, frogs, and mammals), and reported inedible species were not considered for this study. Although mollusks and algae accounted for a significant proportion of fisheries output in live weight, especially in aquaculture, they comprised a very small proportion in edible weight^{15,18,53,54}. Further, we were unable to find publicly available preservation factors (i.e., the ratio of edible portions to final product live weight) for the different processing methods of these groups.”

3. It is a pity that the authors did not consult with the FAO Food Balance Sheets (latest data for 2020) in their analysis as this includes all aquatic food products on a live weight basis, including all mollusks and aquatic plants - these food products would bring down the trophic level of global aquaculture production even further;

The FAO Food Balance Sheets only contain live weight of broad taxonomic groups⁵. There are significant variations in the edible proportions among different mollusk and aquatic plant species, such as Bivalvia, Gastropod, Barnacle, and Ascidiacea. Therefore, if we only have the live weight of mollusks and aquatic plants without more detailed information, the calculated trophic levels of production and consumption would become unreliable.

We agree that the trophic level of global aquaculture production and HATL will further decrease if we combine all aquatic food products including mollusks and aquatic plants. However, as explained in our response to your previous comment, given their very low edible proportions, including these food products would result in excessively low trophic levels in both production and consumption especially for aquaculture. Nonetheless, we have included statements to acknowledge that inclusion of mollusks and aquatic plants would result in a lower trophic level of production and consumption compared to the actual estimated values, which suggests that our estimates of HATL reductions are conservative (L247-251):

“In this study, we did not consider mollusks and aquatic plants. Given that these groups have a lower trophic level than most other aquatic foods and considering the recent increase in the consumption of mollusks and aquatic plants^{2,15,29,44}, our estimate of the current consumption trophic level can be considered conservative (i.e., the inclusion of these groups will likely lower our estimated HATL).”

4. In view of the above, the paper is not a review of global aquatic food consumption patterns, but of selected food products - this should be clearly stated;

Thank you for your comment. We have edited the text in the Introduction to make clear the scope of the study (L79-81):

“We then calculate the HATL and per capita consumption across different countries and regions to analyze global aquatic food (i.e., fish, cephalopods, and crustaceans) consumption patterns, trade characteristics, and impacts.”

We also provide a clear definition of what we mean by aquatic foods in the context of this study in the Methods (L 299-300):

“The term “aquatic foods” is used throughout this study to denote all freshwater and marine fish, cephalopods, and crustaceans.”

5. Statements implying that higher trophic level aquatic species have a better protein quality are not true and should be stated as such (line 67, 74, 149, 152 etc) - the implication being that lower trophic level species have a lower nutritional quality - in fact the reverse is probably true as wild-caught high trophic level species are more likely to be contaminated with environmental contaminants such as POPs, heavy metals, and microplastics;

Thank you for this important comment. We strongly agree and have modified the statements regarding the extended implications of trophic levels to clarify this point (L63-76):

“They not only represent a synthetic metric of species’ diets, which is an important indicator

of different aspects of the environmental footprint of food production for aquaculture and wild caught aquatic foods^{28,29}, but they are also widely recognized as an appropriate indicator of aquatic food value (i.e., higher trophic level generally corresponding to higher price)³⁰⁻³². Although the trophic level of food items in the human diet (human trophic level, HTL) has been considered a simple composite metric that synthetically reflects global patterns of human diet³³, there is currently no quantitative assessment of the human aquatic food trophic level (HATL) and the impacts of trade on it. Nonetheless, we note that some small low-trophic level pelagic and inland fish are also nutrient-rich (e.g., calcium, iron, zinc, long-chain omega-3 polyunsaturated fatty acids)^{6,8,34}, and that wild-capture high-trophic level species are more likely to be contaminated with biomagnifying substances such as persistent organic pollutants (POPs), heavy metals, and microplastics³⁵⁻³⁷. Therefore, the trophic level of aquatic foods can indicate the value of aquatic foods based on price, but it does not predictably reflect the concentration of any nutrients or contaminants status.”

We corrected all the text in the cases where trophic level is used synonymously with nutritional or protein quality (L26, 28, 67, 75, 153, 155, 204, 245, 266).

6. The authors investigate the contribution of aquatic animal food products to total animal protein supply;

Research regarding the contribution of aquatic animal food products to total animal protein supply has been conducted by FAO⁶⁻⁹. Our primary focus lies in analyzing global patterns of aquatic food consumption (live weight availability and HATL), trade characteristics, and impacts.

7. Trade impacts on aquatic food consumption patterns - no mention is given to the costs of aquatic food products and their affordability or not to consumers - in most South American countries poultry is the cheapest source of animal protein whereas in Asia it is the reverse - dried fish usually being the cheapest source of animal protein to the rural poor, including small captured pelagic fish in West African countries;

We thank the reviewer for this important comment. We have edited the relevant discussion in the ‘Discussion’ section to reflect these geographical differences in the access to aquatic food in particular to vulnerable communities (L266-273):

“However, the direct contribution of trade to the food system in vulnerable population groups is limited because the beneficiaries tend to be high-income groups as most exported products consist of high-trophic level species (high-value species) for international markets⁴⁸. Furthermore, differences in dietary habits, income levels, natural resource conditions, and other aspects among different regions can lead to variation in the cost of aquatic products and consumers’ affordability. These situations highlight the need for free, transparent, and adaptive trade and market policies to ensure that all segments of society benefit from international trade⁴⁷.”

8. Finally, the exclusion of the Peruvian anchovy from the analysis (Methods - 263, 289 etc) is not recommended - an increasing proportion being directly consumed as food (it is a food grade product and as such not labeled as forage fish in my opinion), and also have a

low trophic level which would influence final trophic level calculations;

Thank you for your suggestion. We further consulted relevant information about this issue. First, the State of World Fisheries and Aquaculture 2018's Fish utilization and processing section describes how Peruvian anchovy is used as follows:

"Many different species are used for fishmeal and fish oil production, small pelagic species predominating. Many of the species used, such as *Engraulis ringens*, have comparatively high oil yields but are rarely used for direct human consumption."

"In 2016, landings from fisheries directed for fishmeal production were down to less than 15 million tonnes (live weight equivalent) because of reduced catches of *Engraulis ringens*."

Second, IFFO - The Marine Ingredients Organisation also published an article titled "Peruvian Anchovy Why feed, not food?" (<https://www.iffo.com/case-study-peruvian-anchovy-why-feed-not-food>). This article lists a number of reasons why Peruvian anchovies are not used for direct consumption, and the following is a summary of the article.

"Although eaten as whole fish, the majority of Peruvian anchovy are turned into fish oil for feed and capsules, as well as fishmeal, mainly used in aquafeeds. The comparatively low rate of direct human consumption has led some to accuse the industry and the Peruvian Government of depriving local communities of a valuable food source. However, although much effort is, and has been, devoted to promoting the consumption of anchovy in fresh, canned and frozen state, that market remains very small. Despite the efforts of the Peruvian Government and the private sector, direct human consumption of Peruvian anchovy remains at a very low level in the country."

This article also said humans have not been very keen on eating anchovies. Between 2005 and 2011, there was a slight increase in anchovies' consumption, reaching a peak of 1.75% of all anchovies caught in 2011, but it dropped significantly when the purchase obligation for public entities was removed. In recent years, consumption levels have been at half of the 2011 peak.

As we explain in the manuscript (L486-489), we nonetheless could not remove the effect of all other forage species, which to some extent should offset the possible impact in our trophic level calculations from not considering the very small proportion of consumption of the Peruvian anchovy. In any case, we have edited the 'Preprocessing of fisheries datasets' section in the Methods to reflect the reasons behind this choice (L315-319):

"*Engraulis ringens* had comparatively a massive production but is rarely used for direct human consumption in Chile and Peru⁵⁵. Additionally, due to the absence of accurate or even approximate long-term time series data on *Engraulis ringens* consumption proportions, accurate country-level estimations of the theoretical maximum live weight availability for this species were not possible."

Reviewer #2:

This paper connects production with consumption of aquatic foods and explores how trade influenced HATL and consumption patterns. I found the analysis and paper interesting and well written, further increasing our understanding of how trade within the aquatic food system contribute to alleviating nutritional insecurity. As such, it is an important contribution to the scientific community and policymakers. Further, I believe this paper should be accepted for publication, pending some major comments I provide below. These are concerned mostly with the interpretation of the results and conclusions.

We are grateful for your careful review of our manuscript and for your thoughtful and constructive feedback that helped improving our manuscript. Please see below our point-by-point response to your comments. We hope you find our revision satisfying.

The authors used changes in consumption per capita and trophic levels to infer on changes to food security. However, such an inference is only partial. For one, fish's nutritional contribution is more a product of its micronutrient contribution (concentrations) than its protein. The variation in protein concentration among fish (teleost) is very small, but as the authors rightly mentioned at the beginning there are small fish (lower trophic level) with phenomenal micronutrient concentrations. So, I would suggest amending the text in the cases where high trophic level is used synonymously with "higher quality". The importance of tracking trophic levels is well noted but it's difficult to infer on the nutritional quality as a whole just by looking at trophic levels.

Thank you for this important comment, which we strongly agree with.

First, we modified the sentences related to food security, changing "food security changes" to "changes in aquatic food availability" (L191-194):

"Finally, the fact that Africa is notably the only region where both post-trade per capita consumption of aquatic foods and HATL have increased underscores the importance of trade in reducing food deficiency and malnutrition in Africa."

(L264-266):

"Our study supports the conclusion that global trade can improve food availability by allowing most countries to access larger quantities and higher trophic level aquatic foods that otherwise are domestically unavailable."

(L279-282):

"Based on our findings, we believe that aquatic food trade can play substantial and diverse roles in global transformations toward more sustainable and equitable food systems and healthy diets to address multiple forms of food deficiency and malnutrition^{6,8}."

Second, following your suggestion, we have revised the text to avoid making those direct, unidirectional links between trophic level and food quality. We revised the statements regarding the extended implications of trophic levels (L63-76):

"They not only represent a synthetic metric of species' diets, which is an important indicator

of different aspects of the environmental footprint of food production for aquaculture and wild caught aquatic foods^{28,29}, but they are also widely recognized as an appropriate indicator of aquatic food value (i.e., higher trophic level generally corresponding to higher price)³⁰⁻³². Although the trophic level of food items in the human diet (human trophic level, HTL) has been considered a simple composite metric that synthetically reflects global patterns of human diet³³, there is currently no quantitative assessment of the human aquatic food trophic level (HATL) and the impacts of trade on it. Nonetheless, we note that some small low-trophic level pelagic and inland fish are also nutrient-rich (e.g., calcium, iron, zinc, long-chain omega-3 polyunsaturated fatty acids)^{6,8,34}, and that wild-capture high-trophic level species are more likely to be contaminated with biomagnifying substances such as persistent organic pollutants (POPs), heavy metals, and microplastics³⁵⁻³⁷. Therefore, the trophic level of aquatic foods can indicate the value of aquatic foods based on price, but it does not predictably reflect the concentration of any nutrients or contaminants status.”

Third, we corrected the text in the cases where trophic level is used synonymously with nutritional or protein quality (L26, 28, 67, 75, 153, 155, 204, 245, 266).

The authors note positive contributions of trade to consumption per capita amongst most of the population in the world, but looking closely reveals very trivial contributions (Fig 3) for most continents (except North and South America). If the authors would have highlighted the changes to consumption of the ‘before trade’ and the ‘after trade’ in percentages, I would assume we would see very small values in many cases, maybe within the range of uncertainties (which are not noted here). The authors note that >60% of countries experienced an increase in food availability, but how much of those are above, say, 20% change or any other uncertainty level? It is important to note that food availability is only one dimension of food security and other important factors, especially within countries, such as access, affect food security too. Moreover, the authors track food availability, not true consumption at household level. Given the increasing rates of food loss and waste as the industrialization and income of countries increase some of those food gains they noted may be offsetted by supply chain leakages, resulting in smaller gains or perhaps even reduction in actual consumptions. Therefore, the trade analysis offers a glimpse to possible food security gains, but other factors, not addressed here, are as crucial in order to draw a strong conclusion on how trade effects food security. The authors must acknowledge this in their conclusions and limitations more explicitly (not only in the Methods). Perhaps even changing the title of the paper.

Sorry for making this unclear. Indeed, as seen in Figure 3, the impact of aquatic food trade appears negligible for many countries. However, even minor changes in per capita consumption volume resulting from trade are significant for countries with low production and consumption levels. To reflect this better, and after updating population data for certain countries, we recalculated the changes in per capita consumption after trade and found that 66.1% of countries experienced an increase in per capita consumption of aquatic foods (Supplementary Table 2). Next, we calculated the average annual change in per capita consumption of aquatic foods due to trade as a ratio to the average annual per capita production. For reference, we found that only 37.9% of countries had a change rate below

20%, with the majority of countries (62.1%) having a rate exceeding 20%. Nevertheless, we do acknowledge that the precise range of uncertainties in this study cannot be determined due to complex factors. Therefore, we have included reference to the shortage and uncertainties of trade impacts in the new section "Limitations and uncertainties" (L489-495):

"Third, given the increasing rates of food loss and waste linked to the increasing industrialization and income of countries, it is likely that the effects of trade may have been offset to some extent by supply chain leakages, resulting in some cases in smaller gains than those estimated here or perhaps even reduction in actual consumptions. Therefore, our trade analysis provides a baseline for potential food security gains, but we recognize that other factors, not addressed here, are also crucial to draw a strong conclusion on how trade affects food security and should be subject of future studies."

We also agree that availability is just one aspect of food security. We have modified the descriptions related to food security accordingly as explained in our answered to your previous question.

Finally, we have followed your suggestion and changed the title to "Effect of trade on global aquatic food consumption patterns".

Lastly, the authors highlight the aquatic food production and consumption amongst nations, but maybe adding another figure that notes the supply chain flows – i.e., connection of aquatic foods between specific countries (circular flow chart) – can also be revealing in this analysis.

We appreciate your valuable suggestion. Due to data limitations and lack of a precise method, we have not been able to obtain detailed information on the complex trade network relationships in imports and exports of aquatic foods among countries. The most applied database (United Nations Comtrade database) contains bilateral aquatic food trade value information (i.e., data on trade from country A to country B), but does not include quantitative data on trade flows for all aquatic food trade¹⁰. In this study, therefore, we did not focus on trade flows, but we think this is a new very important yet very complex topic. Your suggestion is worthy and we continue to strive building trade flows and supply chains through new method in the process of data collection and analysis. This represents our future direction of effort.

General comments

Line 129 might be necessary to define what trade deficient/surplus means

Agreed. We have added the definitions of trade deficient and surplus to the corresponding section (L132-133).

Fig 2a and b – please make sure the colors of export and import are similar on both subplots

We have changed the colors of export and import subplots to be the same.

Fig 2d,f I would use the same color code as in fig 1 for HATL for consistency and easiness

of comparison.

Corrected. We have changed the colors for all trophic level figures to the same color code.

Line 168: Is “before trade” actually the domestic consumption per capita of what is produced locally? Maybe change the wording to make it clearer.

Thank you. We have revised this sentence to make it clearer (L170-174):

“Differences in per capita consumption of aquatic foods and HATL by country between ‘before trade’ (i.e., consumption stage before trade transactions) and ‘after trade’ (i.e., apparent consumption patterns after completing trade transactions) on reveal the rapidly increasing volume and shifting trade features in various regions that have affected both per capita consumption of aquatic foods and HATL in most parts of the world.”

Line 169 maybe add a sentence that says that you’ve explored the differences between “before” and “after trade” on countries’ consumption per capita and HATL to identify patterns of changes due to trade. It’s not clear enough.

Thank you for your suggestion. We have revised several sentences accordingly (L170-176):

“Differences in per capita consumption of aquatic foods and HATL by country between ‘before trade’ (i.e., consumption stage before trade transactions) and ‘after trade’ (i.e., apparent consumption patterns after completing trade transactions) on reveal the rapidly increasing volume and shifting trade features in various regions that have affected both per capita consumption of aquatic foods and HATL in most parts of the world. Continentally, the per capita consumption of aquatic foods has decreased in Asia and especially South America after trade (Fig. 3a).”

Line 170 unnecessary comma after especially.

Corrected.

Line 172: please write “after/before trade” as a phrase using parenthesis, otherwise its confusing.

Corrected.

Line 235-237 This is inaccurate. Energy cascade among trophic levels of 10% is generally correct, but GHG of wild capture fisheries is dictated by fishing efforts and when it comes to aquaculture with specific management practices, feeding practices (prices, access) and regions, there are exceptions to trophic level being a predictor for environmental impacts. I would cite the Gephart et al, Nature 2021 for the environmental impacts of aquatic foods.

We strongly agree with this comment. Indeed, we can’t directly associate trophic levels with environmental performance, as the latter encompasses multiple dimensions beyond just energy transfer efficiency between trophic levels. The environmental performance of farmed species is also intricately linked to feed inputs and conversion efficiencies which don’t necessarily correspond to trophic level. We also agree with Cottrell, et al. ¹¹ in that the trophic level in aquaculture can no longer be viewed as a trait of the farmed species,

but rather is a dynamic feature of the production system. Consequently, we have interpreted the implications of HATL decline as follows (L252-260):

“Nonetheless, the continuous decrease in the effective trophic level of the majority of farmed species²⁹ coupled with the increasing proportion of low trophic level species in the diet is encouraging for progressing towards reducing dependence on multiple marine ingredients (e.g., fishmeal and oil). Further improvements in resource conversion efficiency on this basis will yield even greater results. Furthermore, aquatic foods not only provide comparatively higher nutrient richness across multiple micronutrients, vitamins, and long-chain polyunsaturated fatty acids relative to terrestrial animal-source foods⁶, but aquatic foods also typically have lower environmental footprint compared to other animal-sourced foods^{45,46}. The observed trends of increasing global contributions from aquatic foods suggest a promising future for more sustainable global diets.”

Line 242-243 only in the past? It's still an on-going question in the present. This present research adds another piece to the puzzle, but it certainly does not resolve this. Trade impacts food security on multiple dimensions, not all of them addressed here.

Thank you for your comments. We have modified this sentence and revised the text related to impacts on food security, narrowing down the implications of this research (please refer to our response to your first two comments).

Line 252 imbalances of what? Not clear

We have modified this sentence to make it clear (L274-275):

“Until now, the problem of imbalances in the growth of demand and aquatic food supply remains prevalent across regions, countries, and income groups^{18,49,50}.”

Reviewer #3:

This paper uses seafood production and trade data from the FAO statistical database and FishBase to develop species-level mass balance data and a trophic level dataset. The authors use the datasets to calculate per capita aquatic food consumption and an aquatic food trophic level at the country level. The authors examine how aquatic food consumption and the aquatic food trophic level is impacted by trade of aquatic food. The work is important given global levels of traded aquatic food and the mixed literature with regards to the impacts of trade on aquatic food security. The results show that the mean aquatic food trophic level has declined. The results also show that international trade has increased aquatic food consumption and the trophic level of most countries as well as reduced geographic differences in trophic levels.

Thank you for your constructive feedback and useful comments. We hope you find our responses satisfying.

The paper is well written, the methodology is sound and described in great detail, and I expect the paper will be of high interest to those in and out of the field. My main concern is related the authors' conclusion that a decline in aquatic trophic level “is potentially good

news from the perspective of the environmental footprint of food production” (line 232-233). The relationship between trophic level and environmental performance is much more nuanced. Gephart et al. (2021, Nature) show that several low trophic level species, like tilapia and shrimp, have higher environmental impacts in many dimensions compared to higher trophic level species like salmon. More so, the environmental performance of aquacultured species is intricately linked to feed inputs and conversion efficiencies which doesn't necessarily correspond to trophic level (Cottrell et al. 2021, Reviews in Aquaculture). I would like to see the authors interpret their results in light of this.

Thank you for your comment. As explained in our response to a similar comment from the previous reviewer, we agree that we can't directly associate trophic levels with environmental performance. The environmental performance of farmed species is also intricately linked to feed inputs and conversion efficiencies which don't necessarily correspond to trophic level. We also agree with Cottrell, et al.¹¹ in that the trophic level in aquaculture can no longer be viewed as a trait of the farmed species, but rather is a dynamic feature of the production system. Consequently, we have interpreted the implications of HATL decline as follows (L252-260):

“Nonetheless, the continuous decrease in the effective trophic level of the majority of farmed species²⁹ coupled with the increasing proportion of low trophic level species in the diet is encouraging for progressing towards reducing dependence on multiple marine ingredients (e.g., fishmeal and oil). Further improvements in resource conversion efficiency on this basis will yield even greater results. Furthermore, aquatic foods not only provide comparatively higher nutrient richness across multiple micronutrients, vitamins, and long-chain polyunsaturated fatty acids relative to terrestrial animal-source foods⁶, but aquatic foods also typically have lower environmental footprint compared to other animal-sourced foods^{45,46}. The observed trends of increasing global contributions from aquatic foods suggest a promising future for more sustainable global diets.”

The authors should also compare the per capita aquatic food consumption estimates in this paper with the FAO food balances and discuss thoughts on why their estimates are in some cases very different from the food balances (e.g., the authors' estimates of per capita consumption for China and Indonesia are much lower).

There are indeed some differences in per capita consumption of aquatic foods for certain countries, which primarily arise from a slight disparity between our equations. We have only calculated the theoretical maximum available volume (i.e., production + imports - exports). FAO's aquatic food supply calculations take also into account the post-balancing non-food uses (Including utilization of aquatic products for reduction to meal and oil, for feed and bait, for ornamental purposes, withdrawals from markets and any other non-food use of fish production) and stock variations. However, FAO acknowledges that information on changes in stocks occurring between the production and the retail levels, or in levels of inventories, is very incomplete³. In most instances data indicated are the minimum required to avoid a negative balance.

Another reason is our exclusive focus on fish, cephalopods, and crustaceans. When we narrow down FAO's data to these categories, the average annual difference in per capita

consumption for China is 1.61 kg/year, while Indonesia's difference is just 0.1 kg/year. When we calculate the average annual per capita consumption relative to FAO's average annual per capita supply, 66.7% of countries have an average change magnitude below 20%, and a substantial majority, 73.6%, fall below the 30% threshold. We also verified the differences between theoretical exports and actual export volume in all 174 countries, which indicate the validity and robustness of our mass balancing method (L456-461).

To summarize all this, we now compare and discuss the discrepancies with FAO Food Balance Sheets in the 'Limitations and uncertainties' section of the Methods, and acknowledge the possible implications and uncertainties resulting from not having accounted for non-food uses and variations in stocks in our balanced data (L480-489):

“Second, to see the magnitude of the differences between both estimates, we compared our balanced average annual per capita consumption to the average annual per capita supply in the FAO Food Balance Sheets. A total of 66.7% of countries had an average difference magnitude below 20%, and a large majority (73.6%) fell below 30%. These differences can mainly be attributed to the fact that the FAO accounts for non-food uses and variations in stocks, as well as differences in live weight conversion factors. Although we removed the world's most important forage fish (i.e., *Engraulis ringens*) from the capture dataset, we couldn't remove the effect of all other existing fish species used in the production of fishmeal, fish oil, and other non-food uses, as well as variations in stocks on aquatic food consumption in all countries.”

References

- 1 Naylor, R. L. *et al.* A 20-year retrospective review of global aquaculture. *Nature* **591**, 551-563, doi:10.1038/s41586-021-03308-6 (2021).
- 2 Naylor, R. L. *et al.* Blue food demand across geographic and temporal scales. *Nat. Commun.* **12**, 1-14, doi:10.1038/s41467-021-25516-4 (2021).
- 3 Fisheries and Aquaculture Software. FishStatJ: Software for Fishery and Aquaculture Statistical Time Series (FAO Fisheries Division, 2023).
- 4 Edwards, P., Zhang, W., Belton, B. & Little, D. C. Misunderstandings, myths and mantras in aquaculture: Its contribution to world food supplies has been systematically over reported. *Marine Policy* **106**, 103547 (2019).
- 5 Fishery and Aquaculture Statistics. Food balance sheets of fish and fishery products 1961-2017 (FishstatJ) (FAO Fisheries Division, Rome, 2020).
- 6 FAO. *The State of World Fisheries and Aquaculture*. Vol. 4 (FAO, 2022).
- 7 FAO. *The State of World Fisheries and Aquaculture*. Vol. 4 (FAO, 2020).
- 8 FAO. *The State of World Fisheries and Aquaculture*. Vol. 4 (FAO, 2018).
- 9 FAO. *The State of World Fisheries and Aquaculture*. Vol. 4 (FAO, 2016).
- 10 Gephart, J. A. & Pace, M. L. Structure and evolution of the global seafood trade network. *Environmental Research Letters* **10**, 125014, doi:10.1088/1748-9326/aae065 (2015).
- 11 Cottrell, R. S. *et al.* Time to rethink trophic levels in aquaculture policy. *Rev. Aquac.*, doi:10.1111/raq.12535 (2021).

Reviewers' Comments:

Reviewer #1:

Remarks to the Author:

All my comments and suggestions have been addressed by the authors

Reviewer #2:

Remarks to the Author:

Dear Authors,

Thank you for adequately addressing all my comments. The paper has indeed improved and I believe it is ready for publication.

Reviewer #3:

Remarks to the Author:

The authors have addressed many of the major questions raised by reviewers, and the manuscript has improved.

Please find some additional comments for further improvement.

There are still indications especially in the results and discussion that changes in HATL infers changes to food security and environmental impacts. The authors should remove the positive and negative connotations when reporting changes in HATL. There is not a clear argument that a higher HATL is better or worse for nutritional or environmental outcomes. For instance, Line 204 states "...trade has improved the availability and trophic level of aquatic foods...". Similarly, line 197 says "trade has positively impacted HATLs.."

There are also some weak, unsupported statements in the discussion. For instance, Line 279, "Based on our findings, we believe that aquatic food trade can play substantial and diverse roles in global transformations toward more sustainable and equitable food systems and healthy diets to address multiple forms of food deficiency and malnutrition." The analysis provides no evidence that trade of aquatic foods increases environmental sustainability, and trade appears to create inequities given that 33% of countries have a decline in per capita consumption after trade.

Also, "the need for free, transparent, and adaptive trade and market policies to ensure that all segments of society benefit from international trade (line 272-273). How will free trade ensure all segments benefit? Free trade tends to result in unequal distribution of wealth.

In its current form, the discussion fails to summarize and highlight the study's contribution to the broader literature and the study's implications.

REVIEWER COMMENTS

NOTE: We address each comment individually below with the original comment in black text and our response in blue text. To help locating our changes in the manuscript, we indicate the lines with “L”.

Reviewer #1:

All my comments and suggestions have been addressed by the authors.

We are grateful for your thorough review.

Reviewer #2:

Thank you for adequately addressing all my comments. The paper has indeed improved and I believe it is ready for publication.

We are grateful for your thorough review.

Reviewer #3:

The authors have addressed many of the major questions raised by reviewers, and the manuscript has improved. Please find some additional comments for further improvement.

We are grateful for your careful review and your thoughtful and constructive feedback that helped improving our manuscript. Please see below our point-by-point response to your comments. We hope you find our revision satisfying.

There are still indications especially in the results and discussion that changes in HATL infers changes to food security and environmental impacts. The authors should remove the positive and negative connotations when reporting changes in HATL. There is not a clear argument that a higher HATL is better or worse for nutritional or environmental outcomes. For instance, Line 204 states “...trade has improved the availability and trophic level of aquatic foods...”. Similarly, line 197 says “trade has positively impacted HATLs.”

Thank you for your comment. We agree that we can’t directly involve the HATL to nutritional or environmental outcomes. We removed the positive and negative connotations when reporting changes in HATL following your suggestion.

(L189-191):

“Finally, the fact that Africa is notably the only region where both post-trade per capita consumption of aquatic foods and HATL have increased underscores the importance of trade in improving aquatic food availability in Africa.”

(L195-196):

“Similarly, trade has also increased HATLs in most countries across continents except Oceania (Fig. 3c).”

(L201-203):

“Globally, international trade has increased the availability and trophic level of aquatic

foods in most (>60%) countries over the past decades (Fig. 3b, d).”

There are also some weak, unsupported statements in the discussion. For instance, Line 279, “Based on our findings, we believe that aquatic food trade can play substantial and diverse roles in global transformations toward more sustainable and equitable food systems and healthy diets to address multiple forms of food deficiency and malnutrition.” The analysis provides no evidence that trade of aquatic foods increases environmentally sustainability, and trade appears to create inequities given that 33% of countries have a decline in per capita consumption after trade.

Thank you. We strongly agree with this comment and have modified the referred statements in the discussion accordingly (L277-284):

“Based on our findings, it is evident that trade has played an important role in harmonizing the global aquatic food consumption. Nevertheless, there are still some countries where per capita consumption of aquatic foods has not improved after trade and remains quite low (Figs. 1e and 3b). Heightened attention and concerted efforts for context-specific mitigation should therefore be given in the future to these countries. To this end, attaining globally equitable trade distribution patterns as well as more harmonized trade environment and policies should be a priority.”

Also, “the need for free, transparent, and adaptive trade and market policies to ensure that all segments of society benefit from international trade (line 272-273). How will free trade ensure all segments benefit? Free trade tends to result in unequal distribution of wealth.

Thank you for your comments. We have modified this sentence to make it more reasonable (L269-271):

“These situations highlight the need for fair, transparent, sustainable and adaptive trade and market policies to ensure that more segments of society benefit from international trade⁴⁷”

In its current form, the discussion fails to summarize and highlight the study's contribution to the broader literature and the study's implications.

Thank you for your comments. We revised the last paragraph in discussion to better summarize and highlight our work's contribution and implications (L284-292):

“Despite the important progress attained in research on aquatic food production and trade, significant challenges persist in achieving a comprehensive understanding of the outcomes of aquatic food trade^{5,17-21}. Our work adds another piece to this puzzle by identifying the implications of trade for contemporary changes in global aquatic food consumption patterns, highlighting the increased availability and trophic level of consumed aquatic foods in a majority of countries with reduced differences in HATL despite the important remaining inequalities. These results provide an important foundation to guide future research on the globalization of aquatic food systems and the impacts of trade on food security.”